# Immediate-Early Modifications to the Metabolomic Profile of the Perilymph Following an Acoustic Trauma in a Sheep Model

**DOI:** 10.3390/jcm11164668

**Published:** 2022-08-10

**Authors:** Luc Boullaud, Hélène Blasco, Eliott Caillaud, Patrick Emond, David Bakhos

**Affiliations:** 1ENT Department and Cervico-Facial Surgery, CHU de Tours, 2 Boulevard Tonnellé, 37044 Tours, France; 2INSERM U1253, iBrain, University of Tours, 10 Boulevard Tonnellé, 37000 Tours, France; 3Department of Biochemistry and Molecular Biology, CHU de Tours, 2 Boulevard Tonnellé, 37044 Tours, France; 4Faculty of Medecine, University of Tours, 10 Boulevard Tonnellé, 37000 Tours, France; 5House Institute Foundation, Los Angeles, CA 90089, USA

**Keywords:** metabolomic, perilymph, acoustic trauma, hearing loss

## Abstract

The pathophysiological mechanisms of noise-induced hearing loss remain unknown. Identifying biomarkers of noise-induced hearing loss may increase the understanding of pathophysiological mechanisms of deafness, allow for a more precise diagnosis, and inform personalized treatment. Emerging techniques such as metabolomics can help to identify these biomarkers. The objective of the present study was to investigate immediate-early changes in the perilymph metabolome following acoustic trauma. Metabolomic analysis was performed using liquid chromatography coupled to mass spectrophotometry to analyze metabolic changes in perilymph associated with noise-induced hearing loss. Sheep (*n* = 6) were exposed to a noise designed to induce substantial hearing loss. Perilymph was collected before and after acoustic trauma. Data were analyzed using univariate analysis and a supervised multivariate analysis based on partial least squares discriminant analysis. A metabolomic analysis showed an abundance of 213 metabolites. Four metabolites were significantly changed following acoustic trauma (Urocanate (*p* = 0.004, FC = 0.48), S-(5’-Adenosyl)-L-Homocysteine (*p* = 0.06, FC = 2.32), Trigonelline (*p* = 0.06, FC = 0.46) and N-Acetyl-L-Leucine (*p* = 0.09, FC = 2.02)). The approach allowed for the identification of new metabolites and metabolic pathways involved with acoustic trauma that were associated with auditory impairment (nerve damage, mechanical destruction, and oxidative stress). The results suggest that metabolomics provides a powerful approach to characterize inner ear metabolites which may lead to identification of new therapies and therapeutic targets.

## 1. Introduction

Hearing loss affects approximately 466 million people (5% of the worldwide population), including 34 million children. Estimates show that by 2050, 900 million people (10% of the worldwide population) will suffer from a disabling hearing loss [1]. The economic cost of hearing loss is estimated to be 750 billion dollars in the USA [1,2]. Hearing loss is the most common sensory deficit and represents a major public health problem [3], negatively affecting oral language development, education, and social interaction [2]. Hearing loss can lead to reduced quality of life by limiting communication and socio-professional relationships; in adults, it leads to social isolation, socio-professional difficulties, and the appearance of a depressive state. Hearing loss has been associated with other public health problems, such as repeated falls, cognitive decline, dementia, and increased hospitalization [3,4]. In about 50% of individuals with sensorineural hearing loss (SNHL), especially post-lingual SNHL, the pathophysiological mechanisms leading to hearing loss and cell loss in the cochlea are myriad and interrelated, and the etiology often remains unknown [5]. Noise-induced hearing loss (NIHL) accounts for 1/3 of SNHL cases, and 10% of the world’s population is exposed to potentially harmful sounds [6].

Several hypotheses have been raised regarding the pathophysiological mechanisms of NIHL. NIHL appears to be the result of a reactive oxygen species (ROS) accumulation in the cochlea due to oxidative stress, mechanical destruction of cells in the auditory nerve organ, and retrocochlear nerve damage [6]. However, the mechanism leading to NIHL is not well understood. Impulse noise can lead to a temporary threshold shift, whereas continuous noise can lead to permanent hearing loss [7,8]. In addition to the cochlear damage that is already known, the impaired neurotransmission associated with the loss of synaptic connections between the inner hair cells and the spiral ganglion neuron fibers, as well as metabolic imbalances, are implicated in elevated hearing thresholds and merit further research to elucidate the mechanisms of hearing loss [9].

Research regarding biomarkers in health has expanded over the past few years, along with an increased interest in omic techniques. Omic techniques comprise genomics (the study of Deoxyribonucleic Acid (DNA)), transcriptomics (the study of Ribonucleic Acids (RNA)), and proteomics and metabolomics. Metabolomics is an emerging technique which can be used to identify biomarkers by analyzing the different metabolites present in a sample. The metabolome refers to all metabolites below 1500 Daltons that provides insight into physiological or pathological states at a given time [10].

To better understand the effects of acoustic trauma on the inner ear, we performed a metabolomic analysis to provide a comprehensive overview of the perilymph fluid metabolome following noise trauma.

## 2. Methods

### 2.1. Experimental Groups and Noise Exposure

All animal procedures were approved by an ethical committee (APAFIS #2018112714344369) and all experiments were performed in accordance with current guidelines and regulations. Six sheep were included in the study (breed = Île-de-France). For the sampling of the perilymph, each animal underwent general anesthesia. General anesthesia was induced by intravenous isoflurane 3% oxygen and 10 mg/kg ketamine along with 0.05 mg/kg xylazine. An oro-tracheal intubation was performed. At the end of the procedure, the sheep were euthanized by overdosing with phenobarbital. Only female sheep (ewes) were used because the cortical bone is thinner than in rams. In each sheep, one ear served as the normal-hearing (NH) model and the contralateral ear was subjected to acoustic trauma and served as the NIHL model; as such, each sheep served as its own comparator. The choice of ear for noise exposure was randomized across sheep. The sound stimulus was 40-ms of pulsatile noise (frequency range = 1–10 kHz) repeated in a loop at an intensity of 120 dB SPL for 1 h. The stimulus was generated by computer using Adobe Audition 2.0^®^ software (Adobe, San Jose, CA, USA), which was connected to an audiometer (Otométrics^®^ Aurical Plus, Natus Medical Incorporated, San Carlos, CA, USA), which was connected to a headset (Peltor^®^, 3M, Saint Paul, MN, USA, H7A, EN352-1:1993). The sound level was measured by a sound level meter (Class 1 Type 2250, Bruel & Kjaer^®^, Naerum, Denmark) and the audiometer was calibrated with white noise. Figure 1 illustrates the study design.

### 2.2. Auditory Measurement

Bone-conduction auditory brain responses (BC-ABRs) were measured to estimate auditory thresholds using a NavPRO ONE bio-logic^®^ Otometrics (Natus Medical Incorporated, San Carlos, CA, USA) setup. BC-ABR thresholds were measured to confirm the NH status at the beginning of the experiment, and then to quantify the hearing loss induced by the acoustic trauma. Auditory thresholds were tested from 50 to 20 dB nHL in 10 dB steps. The stimuli were “clicks” at a modulation frequency of 3000 Hz. Each ear was tested with masking of the contralateral ear by white noise. Thresholds were measured using a B71W bone transducer and transcutaneous needle electrodes. Biolink^®^ software (Advanced Biometric, Mahone, NS, Canada) was used to analyze the response curves, focusing on the latencies and amplitudes of waves I to IV. The NH status was confirmed by the presence of wave IV at 30 dB nHL. As the BC-ABR was measured using bone conduction, measurements cannot be made for intensities >50 dB nHL. In many cases, no wave IV could be observed at 50 dB nHL, indicating that thresholds were higher than the measurement limit. In these cases, 60 dB nHL was considered to be the threshold.

### 2.3. Preparation of the Samples

The perilymph was collected before sound exposure in the NH ear and after sound exposure in the contralateral NIHL ear. The round window was opened with a lumbar puncture needle (22G3 1/2 90 mm 7/10) and the perilymph was collected by capillary action using a micropipette (Microcap 15 μL, length 54 mm, Drummond Scientific Company®, Broomal, PA, USA). All samples were aliquoted into polypropylene tubes and stored at −80 °C until analysis. Metabolite extraction was performed with 400 μL of methanol added to 50 μL of perilymph.

All the samples were shaken for 5 s and incubated at −20 °C for 30 min to deproteinize the sample. After centrifugation that lasted 25 min at 5000× *g* rpm at 4 °C, the supernatant (350 μL) was harvested and evaporated with the SpeedVac concentrator at 40 °C. The dry residue was re-suspended in 100 μL of a methanol/water mixture (75/25) and then 5 μL of sample was injected for liquid chromatography-mass spectrometry analysis. Quality control (QC) samples were prepared by equal volume mixing of all analyzed samples (QC for each sample matrix type).

### 2.4. Metabolomic Analysis

The metabolomic analysis was performed using liquid chromatography coupled to high-resolution mass spectrometry (LC/HRMS) as previously described by our group [11]. The analyses were performed on a UPLC Ultimate WPS-300 system (Dionex, Germany) coupled to a Q-Exactive mass spectrometer (Thermo Fisher Scientific, Bremen, Germany) and the ionization was performed according to the positive (ESI+) and negative (ESI−) electrospray mode [11]. Liquid chromatography was performed with a hydrophilic interaction liquid chromatography (HILIC) column (1.6 μL 150 × 2.10 mm, 100 A), maintained at 40 °C. Two mobile phases were used, and the chromatographic gradients were at a flow rate of 0.3 mL/min. During acquisition, the instrument operated at a resolution of 70,000 (m/z = 200).

Metabolites identification was performed using a standard Mass Spectrometry Metabolite Library compound library (IROA Technologies™, Sea Girt, NJ, USA). Signal values were calculated with Xcalibur^®^ software (Thermo Fisher Scientific, San Jose, CA, USA) by integrating the chromatography peaks corresponding to the selected metabolites. The coefficient of variation (CV) associated with the area of each metabolite was calculated for the control samples (QC) [CV% = (standard deviation/average) × 100]. Metabolites with a CV in the QCs greater than that of the sample and/or CV > 30% were excluded.

### 2.5. Statistical Analysis

In a first step, we evaluated the inter-individual variability of metabolites in perilymph during physiological conditions. Within each sheep, the CV of each metabolite was calculated for the NH ear and compared to the CV for the NIHL ear. Next, we evaluated metabolic changes induced by noise in the perilymph. The percentage of change in relative concentrations of each metabolite in the perilymph between the NIHL and NH ears was calculated and compared to the CV for the same metabolites in QC. These data were analyzed by univariate and multivariate analysis (NH vs. NIHL). All statistical analyses were performed using MetaboAnalyst software (version 5.0; www.metaboanalyst.ca/ (accessed on 25 May 2021)). The univariate analysis of metabolites levels between groups was based on the volcano plot that represents the fold-change (FC) values and the threshold of significance after the non-parametric Wilcoxon test. Significant metabolites were selected by the volcano plot based on FC thresholds < 0.8 or >1.2 and *t*-test *p*-value thresholds < 0.1. Classification was performed by unsupervised Principal Component Analysis (PCA) to visualize the distribution of samples and to highlight putative outsiders. By representing the QC and the different sheep samples, it is possible to understand the inter-individual variability among the sheep in relation to the analytical variability represented by the distribution of the QCs. Partial least squares discriminant analysis (PLS-DA), representing a supervised multivariate analysis, was also performed. The score plot provides an overview of the classified samples. The values of variable influence on projection (VIP) enable the identification of the most important metabolites involved in the discrimination in the supervised multivariate model. The performance of the model was evaluated by the permutation test. Multiple testing was taken into account by the False Discovery Rate (FDR).

### 2.6. Pathways Analyses

Pathways analyses were performed on the most significant metabolites highlighted in the PLS-DA analysis and in univariate analysis. Metabolic pathway enrichment analysis and pathway topology analysis were performed using the MetaboAnalyst computational platform (https://www.metaboanalyst.ca/ (accessed on 25 May 2021)). This strategy presents a single *p* value for each metabolic pathway. Pathway analysis was used to calculate the pathway impact, which represents a combination of the centrality and pathway enrichment results; impact values represent the relative importance of the pathway compared to the others included in the analysis. The pathway impact value was calculated as the sum of the importance measures of the metabolites normalized by the sum of importance measures of all metabolites in each pathway.

An interactive Google-Map-style visualization system was also used to present the analysis results in an intuitive way.

## 3. Results

In this section, we describe the sheep population used, the inducement of NIHL, and the metabolomic results.

### 3.1. Animal Model of Acoustic Trauma

Six sheep (all ewes) were included in the study. The characteristics of the population are presented in Table 1. The average age was 29.9 ± 3 months. The mean weight was 68 ± 11 kg. The average duration of general anesthesia was 205 ± 29.5 min.

Hearing thresholds were compared before and after the acoustic trauma. As shown in Table 2, the mean hearing threshold before noise exposure was 28.3 ± 4.1 dB nHL for the NH ear and 26.7 ± 5.2 dB nHL for the NIHL ear. The mean hearing threshold in the NIHL ear was measured after the acoustic trauma was 58.3 ± 4.1 dB nHL. A paired t-test showed a significant difference in hearing threshold before and after noise exposure [t(5) = 10.3; *p* < 0.001]. All sheep exhibited NIHL after noise exposure.

### 3.2. Effect of Acoustic Trauma on the Perilymph Metabolome

Twelve perilymph samples were collected: six from the NH control ear and six from the NIHL ear. The mean volume of perilymph collected was 89 ± 11 μL. Perilymph from the NH ear was collected at baseline and perilymph from NIHL ear was collected following the noise exposure. The samples were analyzed using LC/HRMS to perform the metabolomic analysis, and 213 metabolites were identified. The PCA score plot based on the metabolic profiles of the QC and sheep samples did not reveal any outliers (Figure 2).

Paired univariate analysis between NH control and NIHL perilymph samples with a *p* < 0.1 and FC ratio < 0.8 or >1.2 revealed four significantly different metabolites: Urocanate (*p* = 0.004; FC = 0.48), S-(5’-Adenosyl)-L-Homocysteine (*p* = 0.060; FC = 2.32), Trigonelline (*p* = 0.060; FC = 0.46), and N-Acetyl-L-Leucine (*p* = 0.090; FC = 2.02).

Paired supervised multivariate analysis showed some discrimination between the NH and NIHL groups; however, this was not significant (accuracy 50%; permutation test *p* = 0.75). This model enabled the identification of metabolites involved in pathophysiological mechanisms associated with acoustic trauma. Some metabolites tended to increase after noise exposure: Urocanate, Oleate, 5-Oxo-L-Proline, N-Acetyl-Glucose, N-Acetylneuraminate, L-Tyrosine, Trigonelline, Leukotriene-B4, 5,6-Dihydrouracil, and 3-Ureidopropionate. Some metabolites tended to decrease after noise exposure: Deoxycarnitine, L-Carnitine, N-Acetyl-L-Leucine, S-(5’-Adenosyl)-L-Homocysteine, and Epinephrine.

### 3.3. Analysis of Metabolic Pathways

Uni- and multi-variate metabolomic analysis of perilymph before and after acoustic trauma revealed significant metabolites of interest. Subsequently, an interpretation of the metabolic pathways involved from these metabolites was performed.

As shown in Table 3, according to all metabolites considered to be relevant after uni- and multi-variate analysis, metabolic pathway analysis revealed the involvement of 5 main pathways: phenylalanine/tyrosine/tryptophan metabolism (*p* = 0.037; impact = 0.5), β-alanine metabolism (*p* = 0.015; impact = 0.16), pantothenate and CoA biosynthesis (*p* = 0.012; impact = 0.05), pyrimidine metabolism (*p* = 0.046; Impact = 0.11), and amino and nucleotide sugar metabolism (*p* = 0.044; impact = 0.05).

An interactive Google-Map visualization system was implemented to facilitate data exploration. Pathway and compound identification was dynamically generated based on interactions with the visualization system (Figure 3). The main metabolic pathway identified was phenylalanine, tyrosine, and tryptophan biosynthesis.

## 4. Discussion

The results of our metabolomic analysis model identified 213 metabolites in sheep perilymph and provide new findings regarding NIHL. Because compounds related to cochlear neurotransmission and metabolic stress were altered by acoustic trauma, this supports the validity of the methodology. Importantly, the analysis identified several previously unknown pathways that were altered by noise exposure (pantothenate and CoA biosynthesis, β-alanine metabolism, and pyrimidine metabolism), indicating that this emerging omic technique may provide new insights regarding metabolome changes in the perilymph triggered by acoustic trauma.

### 4.1. Interest in the Sheep Model

In otology research, rodents have been the most frequently used animal model [12], due to their availability and preferable housing facilities. Yet, rodents show little anatomical, biological, or metabolic similarity to humans with respect to the auditory system [12]. Anatomically, the sheep cochlea has two and a half turns, similar to humans, and has been investigated in a computed tomography study [13]. Unlike the guinea pig, the anatomy of the round window membrane of the sheep shows multiple similarities with humans [14,15]. The auditory spectrum of sheep is comparable to that of humans [16], with humans having an auditory spectrum from 20 to 20,000 Hz and sheep from 100 to 30,000 Hz [16,17]. In rodents and small animals, it is difficult to collect cochlear fluids such as perilymph given the anatomy of the inner ear. In large animals, the perilymph volume is larger and easier to sample through the round window [13,18,19]. For these reasons, sheep may be a better animal model and was therefore used in the present study.

### 4.2. Research Approach

Our research approach appears to be valid and effective. The acoustic trauma technique resulted in significant hearing loss. The sheep were euthanized at the end of the procedure because it is not ethical to wake up a deafened sheep given the damage to the facial nerve. As such, we did not re-measure thresholds to observe whether any hearing recuperation occurred. Two types of noise stimuli can be used to induce acoustic trauma: impulse noise and continuous noise. Impulse noise causes immediate damage and a shift in transient thresholds, whereas continuous noise causes cochlear damage with inflammatory healing, resulting in transient and permanent hearing loss over the longer term [7]. The choice of impulse noise was dictated by the objective of causing rapid lesions with significant hearing loss in order to observe consequences at the metabolic level. The choice of 120 dB SPL stimulus intensity with 1 h of exposure was motivated by the fact that synaptopathy lesions are visible from a threshold of 100 dB SPL [18] and cochlear lesions can be observed from a threshold of 115 dB SPL [20]. The impulse noise of 120 dB SPL for 1 h resulted in substantial and significant hearing loss.

### 4.3. Metabolomics as a Powerful Approach to Characterize Perilymph

To date, six previous studies on inner ear metabolomics have been published [18,19,21,22,23,24]. Two studies focused on the metabolomic analysis of perilymph during cochlear implantation in humans [21,22]. One study analyzed perilymph fluid following cisplatin-induced ototoxicity in guinea pigs [24]. For an analysis of the metabolomics of the perilymph following NIHL, two studies were performed using guinea pigs and found approximately 100 metabolites [19,23], far fewer than the 213 metabolites found in the present study. Another study analyzed metabolomics following NIHL using mice [18], finding 220 metabolites [18]; however, the whole inner ear was sampled (bone and perilymph), as opposed to only perilymph in the present study. The abundance of metabolites found in the present study may be attributed to metabolomic analysis technique [21], the larger volume of perilymph collected, and the use of capillarity to sample the perilymph.

### 4.4. Metabolomic Modifications following Acoustic Trauma

Our results suggested the implication of different pathways to explain the pathophysiological mechanisms following acoustic trauma. Indeed, following acoustic trauma, our hypotheses are mechanical destruction of hair cells, damages of the synapses and nerve, and oxidative stress due to inflammatory reactions.

#### 4.4.1. Neurotransmission

Of particular interest is our finding is that noise had an impact on the biosynthesis of phenylalanine, tyrosine, and tryptophan that are aromatic amino acids, the precursors of the monoamine neurotransmitters, serotonin, and catecholamines (dopamine, norepinephrine, and epinephrine) [25]. This point was raised in a previous study where alterations in these amino acids were also observed in the mammalian cochlea [18]. One study has shown that aromatic amino acids suppress discharge by afferent fibers innervating hair cells [26]. Their exact function in NIHL remains to be demonstrated.

#### 4.4.2. Oxidative Stress

The oxidative stress pathway following acoustic trauma and its counterbalance by the action of antioxidant systems are well documented [27] and are reflected in our results. First, urocanate was significantly increased after acoustic trauma. It is likely to lead to the production of ROS responsible for increased cell death [27]. This has been observed in the brain, where urocanate has been associated with an increase in glutamate synthesis [28]. An excessive release of glutamate and/or hyper activation of glutamate receptors in the cochlea results in excitotoxicity responsible for damage to hair cell synapses [9]. Our results suggest the same mechanism in the perilymph.

Our findings also suggest the involvement of the pantothenate and CoA biosynthetic pathway after sound exposure, similar to other animal studies [23]. CoA is a cofactor for a multitude of enzymatic reactions, including the oxidation of fatty acids, carbohydrates, pyruvate, lactate, ketone bodies, and amino acids [29]. Pantothenate can regulate CoA synthesis in cell membranes and protect against increased oxidative stress by reducing the level of malondialdehyde (MDA), the major product of lipid peroxidation. It can inhibit the inflammatory process by reducing the level of inflammatory reactive proteins and promoting CoA and glutathione levels [30]. An implication of S-(5’-Adenosyl)-L-Homocysteine was found in the NIHL model. Partearroyo et al. [31] studied homocysteine and hearing loss and found that homocysteine is involved in oxidative stress.

#### 4.4.3. Mechanical Destruction

The present results suggest the mechanical destruction of cells following acoustic trauma according to the presence of ß-alanine and N-acetylneuraminate metabolites. ß-alanine is present in hair cells; its increase in the perilymph following noise exposure suggests cellular destruction [32]. N-acetylneuraminate, a derivative of sialic acid, is usually located on the terminal portion of glycoproteins and glycolipids located on the surface of the cell membrane [22]. High concentrations of this metabolite can be explained by the rupture of the cell membrane and are found during cell death and the onset of apoptosis.

#### 4.4.4. Nerve Damage

The present results suggest nerve damage occurred, given the difference in perilymph composition in trigonelline and N-Acetyl-L-Leucine between the NH and NIHL ears. Trigonelline was significantly altered following acoustic trauma. Its role has been demonstrated in the reduction in auditory damage and in hearing loss in mice [33,34]. Trigonelline can also reduce neurodegenerative effects with the use of nerve growth factor. It reduces oxidative stress and causes neuroprotective effects and interacts with ototoxic signaling pathways [33,34]. These results are of particular interest in terms of associations between trigonelline and NIHL; to date, literature on this subject is non-existent.

We observed a decrease in N-Acetyl-L-Leucine, which is mainly known for its vestibular action. It acts by normalizing the membrane potential at the level of the vestibular nuclei, allowing for improvement in balance function, and affects the regrowth of neurons. The consumption of N-Acetyl-L-Leucine in the perilymph following acoustic trauma may be explained by its effect on nerve healing [35]. To date, there are no studies on NIHL and N-acetyl-L-leucine.

### 4.5. Limits to the Study

The small number of sheep tested is a limitation to this study. However, this study represents important preliminary work for future metabolomic analysis studies in larger populations.

The duration of anesthesia is another limitation of this study. The perilymph metabolome results could have been impacted by the drugs used and the duration of procedure [36]. However, considering the pharmacokinetic properties of the anesthesia medications used, this seems unlikely. Likewise, no link has been established in the literature between the anesthesia medications used and metabolites. An alternative approach would have been to randomize the perilymph sampling across ears and sheep (e.g., first sampling the perilymph in the NIHL ear after acoustic trauma and then measuring thresholds and sampling perilymph in the NH control ear). However, this approach may have induced contralateral hearing loss in the NH ear, thereby distorting the results.

It would have been interesting to know whether hearing thresholds recovered after acoustic trauma and if so, the time course of the recovery. However, this extended testing was deemed to be unethical in consideration of the welfare of the animals.

## 5. Conclusions

These preliminary results suggest that in case of NIHL, several metabolic pathways are involved that relate to mechanical destruction, oxidative stress, neurotransmission, and nerve damage. Metabolomic analysis is still in the early stages of development as an approach to study SNHL. The identification of specific metabolites as biomarkers or a metabolomic profile for NIHL could be used to relate hearing loss with noise exposure, especially in cases where the cause of hearing loss is unknown. Metabolomic analysis may also contribute to development of therapeutics for NIHL. Further studies are needed to confirm these results and to develop an atraumatic approach to sample the perilymph, especially in humans.

## Figures and Tables

**Figure 1 jcm-11-04668-f001:**
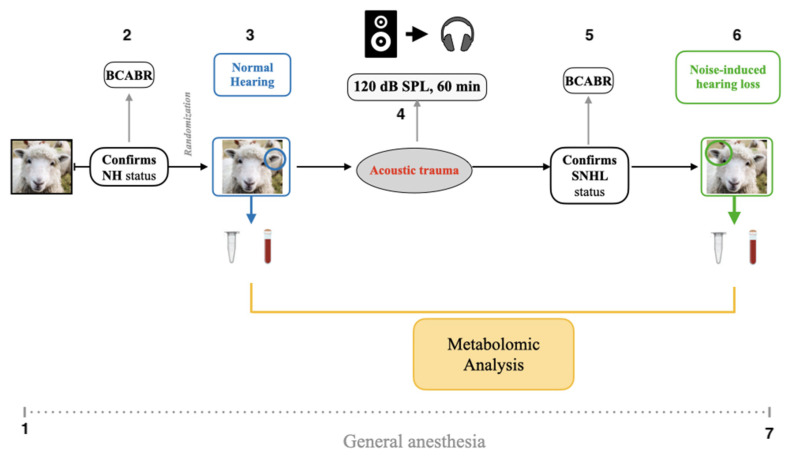
Study design: (1) Sheep under general anesthesia. (2) Perform BC-ABR to confirm NH status in NH ear. (3) Sample perilymph from NH ear. (4) 60 min of impulse noise exposure at 120 dB SPL. (5) Perform BC-ABR to confirm SNHL status in NIHL ear. (6) Sample perilymph from NIHL ear. (7) Sheep euthanized. The polyethylene tube represents the perilymph sample, blood tube represents the serum.

**Figure 2 jcm-11-04668-f002:**
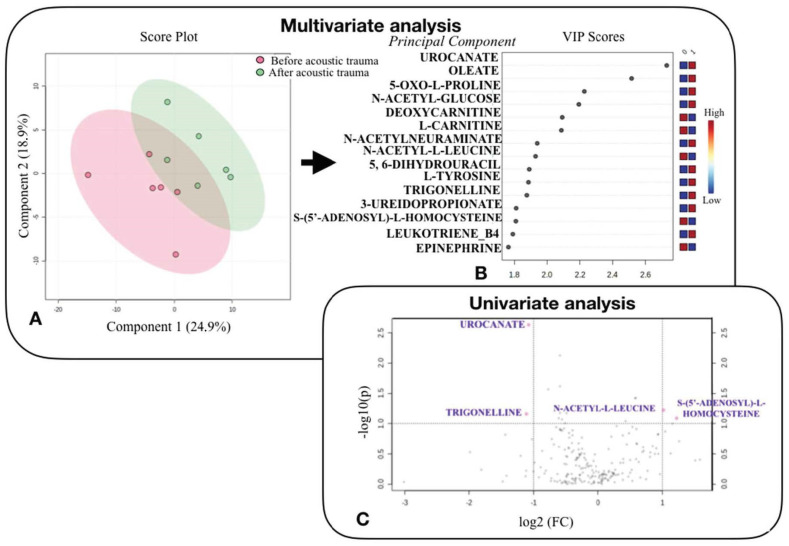
Statistical analysis to compare the metabolomic profile of perilymph fluid from sheep before and after acoustic trauma. (**A**): Multivariate analysis using partial least squares discriminant analysis (PLSDA) to distinguish metabolomic profile of perilymph fluid from sheep before (red circles) and after acoustic trauma (green circles). Components 1 and 2 represent a linear combination of relevant metabolites expressing maximum variance. (**B**): The rank of different metabolites (top 15) identified by PLSDA according to the VIP (Variable Influence of Projection) score on the left. The colored boxes on the right indicate the relative concentrations of the corresponding metabolite in each group studied. (**C**): Univariate analysis via a plot based on fold change and *p*-value, highlighting 4 metabolites. The volcano plot based on the comparison between NH control and NIHL perilymph samples, highlighting metabolites characterized by a FC > 1.2 concentration ratio and a t-test (y) < 0.1 (pink points). Note that fold changes and *p*-values are log transformed. The further away from (0.0) position, the more important the feature.

**Figure 3 jcm-11-04668-f003:**
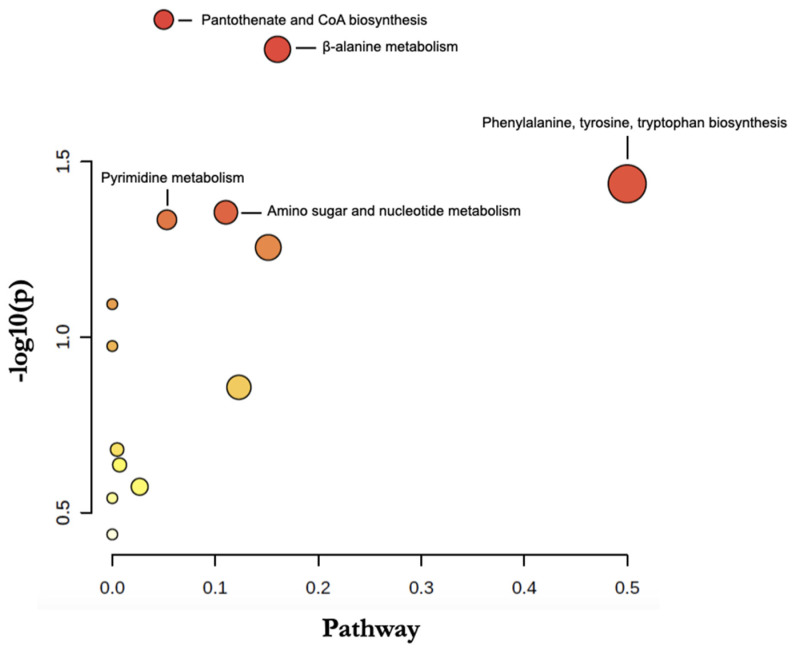
Metabolomic Pathway Analysis based on the 15 VIP metabolites the most discriminant in the multivariate model explaining hearing loss. The circles resume all matched pathways according to the *p* values from the pathway enrichment analysis and pathway impact values from the pathway topology analysis (calculated as the sum of importance measures of the matched metabolites normalized by the sum of the importance measures of all metabolites in each pathway). Each circle represents a metabolite set with its color based on its *p*-value (darker colors indicate more significant changes in metabolites belonging to the corresponding pathway), and its size is based on pathway impact score. The most impacted pathways are annotated.

**Table 1 jcm-11-04668-t001:** Characteristics of the 6 included sheep.

Sheep	Breed	Sex	Age (Months)	Weight (kg)	Duration of Anesthesia (Min)	Stimulation Sampling Time (Min)
1	IDF	F	27.5	64	240	45
2	IDF	F	27.8	52	240	80
3	IDF	F	28.4	72	180	60
4	IDF	F	30.0	82	180	50
5	IDF	F	35.5	63	210	60
6	IDF	F	29.9	72	180	45
Mean			29.9	67.5	205	56.7
SD			2.97	10.23	29.5	13.3

Legend: SD: Standard deviation, IDF: Ile-De-France, F: female.

**Table 2 jcm-11-04668-t002:** Auditory thresholds (wave IV from BC-ABRs) for the NH model ear before perilymph sampling, and for the NIHL model ear before perilymph sampling and after acoustic trauma.

	NH Model		NIHL Model		
Sheep N°	Side	Threshold before Sampling (dB)	Side	Threshold before Sampling (dB)	Threshold after Acoustic Trauma (dB)
1	L	30	R	30	60
2	L	30	R	30	50
3	R	30	L	30	60
4	R	30	L	30	60
5	R	30	L	20	60
6	L	20	R	20	60
Mean	-	28.3		26.7	58.3
SD	-	4.1		5.2	4.1

NH, normal hearing; NIHL, noise-induced hearing loss. Legend: SD: Standard Deviation, NIHL: noise-induced hearing loss, L: Left, R: Right, dB: decibel.

**Table 3 jcm-11-04668-t003:** Significant metabolic pathways involved in noise-induced hearing loss via perilymph analysis. The *p*-value was calculated from the enrichment analysis. *p*-FDR is the adjusted *p*-value using the false discovery rate. Impact values were calculated from the path topology analysis.

Pathway	*p*-Value	*p*-FDR	Impact
Pantothenate and CoA biosynthesis	0.012	0.64	0.05
β-alanine metabolism	0.015	0.64	0.16
Phenylalanine, tyrosine, tryptophan biosynthesis	0.037	0.78	0.50
Amino sugar and nucleotide metabolism	0.044	0.78	0.11
Pyrimidine metabolism	0.046	0.78	0.05

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
