# Peer review of "Immediate-Early Modifications to the Metabolomic Profile of the Perilymph Following an Acoustic Trauma in a Sheep Model"

_jcm, 2022, doi:10.3390/jcm11164668_

Round 1

Reviewer 1 Report

The pathophysiology of sensorineural hearing loss due to noise is unclear, and the incidence is high. This study is an exceptional scientific contribution to the understanding of metabolic changes in the perilymph after noise exposure, by introducing a new diagnostic method - metabolomic analysis, which enables a new approach to the perception and analysis of sensorineural hearing loss. By identifying different metabolic profiles of perilymph and metabolic pathways associated with sensorineural hearing loss, this study provides an explanation of the pathophysiological basis of acoustic trauma through different metabolic disorders, and thus opens the door to new insights not only into pathological changes in perilymph due to acoustic trauma  and modalities that could be applied in their treatment. The study is extremely well designed, documented, with results that are impressive with a discussion that is a step into the future.

Author Response

Reviewer 1 :

Accepted, no revisions needed

->Many thanks for your evaluation

Reviewer 2 Report

The submitted manuscript is interesting; the topic of the study is innovative, and the model used is unusual but well justified. The authors present the results of their study on early changes in the perilymph metabolome during noise exposure.

Materials and Methods

Figure 1 mentions that the animals were under general anesthesia. What was the anesthetic used? How was the anesthesia applied (i.v.; i.p.; s.c.; inhalation?)? Was it applied only once or continuously?

How long in total (please provide the mean of time and the range) were the animals under anesthesia?

Please add this information in 2.1

Please add the gender information to Table 1 (were rams, ewes, or both used?).

Please substitute “acoustic shock” with “noise exposure” or “acoustic trauma”.

Line 229-230 – “Multivariate analysis using partial least squares discriminant analysis (PLSDA) to distinguish ewes before or after sound exposure.” – to distinguish ewes? Please clarify.

The experimental design did not take into account the effect of anesthesia. The animals spent at least 120 minutes under anesthesia. The control samples were taken at the beginning of that period, whereas “noise-exposed” samples were taken at the end. That should be listed as one of the shortcomings of this work.

I suggest mentioning in the title and abstract that this study focused on immediate-early changes in the metabolome of fluids from scala tympani following noise exposure.

It would also be worth discussing (at least hypothetically) how the metabolome changes could occur in the scala tympani within the 120 minutes (noise exposure and ABR measurements). Please add the information about how the compounds identified by the study could be catabolized and in which pathways. Also, please discuss and cite the work that studied metabolic processes in the cochlea combined with hearing loss. See https://doi.org/10.3389/fnmol.2017.00107 for some hints. 

Author Response

Reviewer 2 :

The submitted manuscript is interesting; the topic of the study is innovative, and the model used is unusual but well justified. The authors present the results of their study on early changes in the perilymph metabolome during noise exposure.

Materials and Methods

Figure 1 mentions that the animals were under general anesthesia. What was the anesthetic used? How was the anesthesia applied (i.v.; i.p.; s.c.; inhalation?)? Was it applied only once or continuously?

->We have clarified: “For the sampling of the perilymph, each animal underwent general anesthesia. General anesthesia was induced by intravenous Isofluorane 3% oxygen and 10mg/kg Ketamine along with 0.05 mg/kg Xylazine. An oro-tracheal intubation was performed. At the end of the procedure, the sheep were euthanized by overdosing with Phenobarbital.”

How long in total (please provide the mean of time and the range) were the animals under anesthesia? Please add this information in 2.1

->We have clarified in the Results: “The average duration of general anesthesia was 205 ± 29.5 minutes."

Please add the gender information to Table 1 (were rams, ewes, or both used?).

->We have clarified: “Only female sheep (ewes) were used because the cortical bone is thinner than in rams."

Please substitute “acoustic shock” with “noise exposure” or “acoustic trauma”.

->We have replaced “acoustic shock” with “acoustic trauma” throughout the MS.

Line 229-230 – “Multivariate analysis using partial least squares discriminant analysis (PLSDA) to distinguish ewes before or after sound exposure.” – to distinguish ewes? Please clarify.

->We have revised the Figure 2 legend as: “Statistical analysis to compare the metabolomic profile of perilymph fluid from sheep before and after acoustic trauma. A: Multivariate analysis using partial least squares discriminant analysis (PLSDA) to distinguish metabolomic profile of perilymph fluid from sheep before (red circles) and after acoustic trauma (green circles). Components 1 and 2 represent a linear combination of relevant metabolites expressing maximum variance. B: The rank of different metabolites (top 15) identified by PLSDA according to the VIP (Variable Influence of Projection) score on the left. The colored boxes on the right indicate the relative concentrations of the corresponding metabolite in each group studied. C: Univariate analysis via a plot based on fold change and p-value, highlighting 4 metabolites. The volcano plot based on the comparison between NH control and NIHL perilymph samples, highlighting metabolites characterized by a FC>1.2 concentration ratio and a t-test (y) <0.1 (pink points). Note that fold changes and p-values are log transformed. The further away from (0,0) position, the more important the feature.”

The experimental design did not take into account the effect of anesthesia. The animals spent at least 120 minutes under anesthesia. The control samples were taken at the beginning of that period, whereas “noise-exposed” samples were taken at the end. That should be listed as one of the shortcomings of this work.

->Thank you for your comment. We have added this limitation to the Discussion: “The duration of anesthesia is another limitation of this study. The perilymph metabolome results could have been impacted by the drugs used and the duration of procedure [36]. But considering the pharmacokinetic properties of the anesthesia medications used, this seems unlikely. Also, no link has been established in the literature between the anesthesia medications used and metabolites. An alternative approach would have been to randomize the perilymph sampling across ears and sheep (e.g., first sampling the perilymph in the NIHL ear after acoustic trauma and then measuring thresholds and sampling perilymph in the NH control ear). However, this approach may have induced contralateral hearing loss in the NH ear, thereby distorting the results.”

I suggest mentioning in the title and abstract that this study focused on immediate-early changes in the metabolome of fluids from scala tympani following noise exposure.

->We have modified the title as: “Immediate-early modifications to the metabolomic profile of the perilymph following an acoustic trauma in a sheep model.” We have modified the abstract to include: “The objective of the present study was to investigate immediate-early changes in perilymph metabolome following acoustic trauma.”

It would also be worth discussing (at least hypothetically) how the metabolome changes could occur in the scala tympani within the 120 minutes (noise exposure and ABR measurements). Please add the information about how the compounds identified by the study could be catabolized and in which pathways. Also, please discuss and cite the work that studied metabolic processes in the cochlea combined with hearing loss. See https://doi.org/10.3389/fnmol.2017.00107 for some hints.

->We have added to the Discussion: “An implication of S-(5'-Adenosyl)-L-Homocysteine was found in the NIHL model. Partearroyo et al. [31] studied homocysteine and hearing loss and found that homocysteine is involved in oxidative stress. »

Reviewer 3 Report

The aurors present a metabolomic analysis to provide a comprehensive overview of the perilymph fluid metabolome following noise trauma.

While there is an overall good description on the potential of the metabolomics research in the field, the paper should be improved in terms of presenting its results and the merits it offers. Conclusion is poorly written and too generic for the results achieved, so this should be rewritten.

There are some other 'grey' areas throughout the paper that need further clarification. A key point for example, and what I miss is a link to the Ethical Committee's approval for the experiments, as there is no link to it mentioned (p2, l.76-77). Another issue not described relates to the health condition of the sheep used, where the word 'sacrifice' used in p.3, l93 can be perceived wrong. As it is indicated later (p.9 l305-307) the sheep were euthanised, so that should be depicted correctly.

The paper could also benefit from an overall english grammar and syntactic proofreading, as there are several issues identified. To name some:
p1, l.43 "negatively affecting can affect"

p2., l.64 "since a few years"

p4, l.153 "software Metaboanalyst sofware"

p4, l.171 "Pathways" instead of "Pathway" used in the 2.6 Heading

Thank you for taking these points under consideration

Author Response

Reviewer 3 :

The authors present a metabolomic analysis to provide a comprehensive overview of the perilymph fluid metabolome following noise trauma. While there is an overall good description on the potential of the metabolomics research in the field, the paper should be improved in terms of presenting its results and the merits it offers. Conclusion is poorly written and too generic for the results achieved, so this should be rewritten.

->We have revised the MS to better present the Results. We have also revised the Conclusion as: “These preliminary results suggest that in case of NIHL, several metabolic pathways are involved that relate to mechanical destruction, oxidative stress, neurotransmission and nerve damage. Metabolomic analysis is still in the early stages of development as an approach to study SNHL. Identification of specific metabolites as biomarkers or metabolomic profile for NIHL could be used to relate hearing loss with noise exposure, especially in cases where the cause of hearing loss is unknown. Metabolomic analysis may also contribute to development of therapeutics for NIHL. Further studies are needed to confirm these results and to develop an atraumatic approach to sample the perilymph, especially in humans.”

There are some other 'grey' areas throughout the paper that need further clarification. A key point for example, and what I miss is a link to the Ethical Committee's approval for the experiments, as there is no link to it mentioned (p2, l.76-77).

->No link is provided by ethical committee in France. We state in the beginning of the Methods: “All animal procedures were approved by an ethical committee (APAFIS #2018112714344369) and all experiments were performed in accordance with current guidelines and regulations.” We have also added at the end of the MS: “Institutional Review Board Statement: The animal study protocol was approved by the Ethics Committee of CEEA Val de Loire N°19 (APAFIS #2018112714344369 and 21/03/2019) for studies involving animals.”

Another issue not described relates to the health condition of the sheep used, where the word 'sacrifice' used in p.3, l93 can be perceived wrong. As it is indicated later (p.9 l305-307) the sheep were euthanised, so that should be depicted correctly.

->We have removed the word “sacrificed” throughout the MS and use the word “euthanized” where appropriate.

The paper could also benefit from an overall english grammar and syntactic proofreading, as there are several issues identified.

->The manuscript has been edited for language by Dr John Galvin (House Institute Foundation, Los Angeles, USA).

To name some:

p1, l.43 "negatively affecting can affect"

->Corrected

p2., l.64 "since a few years"

->Revised as: “Research regarding biomarkers in health has expanded over the past few years, along with increased interest in omic techniques.”

p4, l.153 "software Metaboanalyst sofware"

->Corrected

p4, l.171 "Pathways" instead of "Pathway" used in the 2.6 Heading

->Corrected

Round 2

Reviewer 3 Report

Thank you for taking under consideration my concerns and comments, I really appreciate it!